# In Vitro Effects of Bisphenol A and Tetrabromobisphenol A on Cell Viability and Reproduction-Related Gene Expression in Pituitaries from Sexually Maturing Atlantic Cod (*Gadus morhua* L.)

**Kristine von Krogh** [1] , **Erik Ropstad** [2] , **Rasoul Nourizadeh-Lillabadi** [1] , **Trude Marie Haug** [3] and **Finn-Arne Weltzien** [1,*]

1. Department of Basic Sciences and Aquatic Medicine, Faculty of Veterinary Medicine, Norwegian University of Life Sciences, 0454 Oslo, Norway
2. Department of Production Animal Clinical Sciences, Faculty of Veterinary Medicine, Norwegian University of Life Sciences, 0454 Oslo, Norway
3. Department of Oral Biology, Faculty of Dentistry, University of Oslo, 0316 Oslo, Norway
* Correspondence: finn-arne.weltzien@nmbu.no; Tel.: +47-67232036

**Abstract:** Bisphenol A (BPA) and tetrabromobisphenol A (TBBPA) are widely used industrial chemicals, ubiquitously present in the environment. While BPA is a well-known endocrine disruptor and able to affect all levels of the teleost reproductive axis, information regarding TBBPA on this subject is very limited. Using primary cultures from Atlantic cod (*Gadus morhua*), the present study was aimed at investigating potential direct effects of acute (72 h) BPA and TBBPA exposure on cell viability and the expression of reproductive-relevant genes in the pituitary. The results revealed that both bisphenols stimulate cell viability in terms of metabolic activity and membrane integrity at environmentally relevant concentrations. BPA had no direct effects on gonadotropin gene expression, but enhanced the expression of gonadotropin-releasing hormone (GnRH) receptor 2a, the main gonadotropin modulator in Atlantic cod. In contrast, TBBPA increased gonadotropin transcript levels but had no effect on GnRH receptor mRNA. In conclusion, both anthropogenic compounds display endocrine disruptive properties and are able to directly interfere with gene expression related to reproductive function in cod pituitary cells at environmentally relevant concentrations in vitro.

**Keywords:** gonadotropin; GnRH receptor; endocrine disruption; reproduction; teleost

## 1. Introduction

During the last few decades, attention towards the potentially harmful effects of industrial chemicals released into the environment has increased dramatically. Many of these chemicals are endocrine disruptive compounds (EDCs), capable of interfering with normal endocrine function in animals, including fish. One important and susceptible endocrine system is that of reproduction. The physiological compartments of vertebrate reproduction comprise of the brain—pituitary—gonadal (BPG) axis, where neuroendocrine and endocrine substances relay communication between the different axis levels. In teleosts, gonadotropin-releasing hormone (GnRH), released through neuron fibers from the preoptic area and binding to its receptors in the pituitary, stimulates gonadotropin synthesis and release [1]. The gonadotropins, follicle-stimulating hormone (FSH) and luteinizing hormone (LH), in turn, bind to receptors in the gonads, stimulating gametogenesis and steroidogenesis. Once in the bloodstream, sex steroids will provide feedback to the higher levels of the BPG axis [2–4].



Bisphenol A (BPA) and its halogenated derivative tetrabromobisphenol A (TBBPA) are widely used industrial and commercial chemicals, the former mainly used in the production of polycarbonate plastics and epoxy resins, and the latter used as a flame retardant in combustible products [5,6]. Both bisphenols are ubiquitously detected in the environment, although at highly variable concentrations dependent on sample matrix and location [7–12]. Peak measurements of BPA have been 21 µg/L in surface waters, 56 µg/kg dry weight (DW) in suspended solids and 17.2 mg/L in landfill leachates [7–9]. Seawater levels are usually below 1 µg/L, but BPA could potentially be leaching at marine sites of accumulated plastic waste. Further, BPA leaches faster and may withstand degradation longer in saltwater than in freshwater [7,8,13–16]. Measured peak levels of TBBPA have been 4.87 µg/L in lake waters, 600 µg/kg DW in sewage sludge, 9.8 mg/kg DW in freshwater sediment and 1.8 µg/L in seawater [10–12].

BPA is a well-known endocrine disruptor and has been demonstrated to act on all levels of the teleost BPG-axis; brain [17–21], pituitary [22–25], and gonads [26–29]. Considerably less information is currently available for TBBPA in this regard, with only a few teleost reproductive studies conducted. These studies indicate that TBBPA can affect teleost reproductive success through decreased egg production, delayed hatching, and decreased hatching rate [28,30,31]. Estrogenic activity from TBBPA exposure has been demonstrated in mammalian in vitro systems [32–34], while in teleosts, estrogenic activity has, to our knowledge, not been studied in vitro, and results in vivo are conflicting [30,35–39]. However, the capacity of TBBPA to act as a thyroid-disrupting compound has clearly been demonstrated in both mammalian and teleost assays, e.g., TBBPA can bind transthyretin with higher potency than the natural ligand thyroxine [30,33,40–42].

For both BPA and TBBPA, the vast majority of reproductive studies have been on gonads, with fewer studies focusing on the higher BPG-axis levels. For TBBPA, no such studies exist in teleosts, while to our knowledge only two studies exist for other vertebrate taxa [43,44]. While BPA is rather well-studied, most teleost reports are from small, freshwater species. The aim of the present study was therefore to investigate potential direct effects of BPA and TBBPA at the pituitary level in a marine teleost, the Atlantic cod (*Gadus morhua*). Using pituitary primary cultures from sexually maturing individuals, cell viability and expression of reproductive related genes, i.e., the gonadotropins (*FSHb* and *LHb*) and two gonadotropic GnRH receptors (*GnRHr1b* and *GnRHr2a*), following acute exposure to a range of concentrations were assessed.

## 2. Materials and Methods

### 2.1. Animals

Atlantic cod of both sexes were captured from the southern Norwegian coast (approx. 60° N) on four separate occasions. Specific ethical approval for this study was not needed as the animals themselves were not experimentally treated (Norwegian legislation for use of animals in research, Chapter II, §6). Care was nonetheless taken so that stress and suffering of the animals were minimized. All fish were euthanized by swift decapitation at location prior to immediate dissection of pituitary and gonads. Dissected pituitaries were kept in modified L-15 medium (Life Technologies, Carlsbad, CA, USA, see below for modifications) on ice until culture preparation. After dissection, body and gonads were weighted and the gonadosomatic index (GSI; (Gonadal weight/Total body weight) × 100) calculated. As GSI by itself is not a precise indicator of sexual maturity, gonads were also visually inspected and staged according to von Krogh et al. [3] (Supplementary data). In total, four cultures were prepared using pooled pituitaries from sexually maturing donor fish (*n* = 48; body weight: 1.48 ± SD 0.56 kg; GSI: 6.35 ± SD 3.28). Sex ratios of donors within cultures were 3M/3F, 3M/1F, 11M/6F, and 10 M/11F (M: males; F: females).

### 2.2. Dispersed Pituitary Primary Cell Cultures

The present study followed a previously described, optimized protocol for primary cultures of Atlantic cod pituitaries [45]. The optimized conditions included cell density, $pCO_2$, and incubation

temperature, as well as pH and osmolality for working solutions. In short, pooled pituitaries were chemically and mechanically dissociated and seeded in culture wells pre-coated with poly-L-lysine (0.1 mg/mL, Sigma-Aldrich, St. Louis, MO, USA) at a density of $1.5 \times 10^5$ cells/cm$^2$. Cells were incubated in modified L-15 medium, adjusted to 320 mOsm, at 12 °C in a humidified atmosphere of 0.5% $CO_2$ (p$CO_2$; 3.8 mmHg, which resulted in a medium pH of 7.85). After 24 h, culture medium was replaced to remove damaged and detached cells.

### 2.3. BPA and TBBPA Exposure

Stock solutions of bisphenol A (BPA; 4,4′-(propane-2,2-diyl)diphenol, Sigma-Aldrich, CAS: 80-05-7, ≥99%) and tetrabromobisphenol A (TBBPA; 4,4′-(propane-2,2-diyl)bis(2,6-dibromophenol), Sigma-Aldrich, CAS: 79-94-7, 97%) were prepared in 100% ethanol (EtOH; Kemetyl, Kolbotn, Norway) and stored at −20 °C for up to one week. Prior to cell exposure, stocks were diluted to the desired concentration in modified L-15 medium, with working solutions having a final EtOH concentration of 0.2% (34.25 mM). For each experiment, controls w/wo EtOH (solvent control/control blank) were included. For viability tests, exposure doses were $10^{-9}$ to $10^{-3}$ M (0.22 to 228,290 µg/L) for BPA and $10^{-9}$ to $10^{-4}$ M (0.54 to 54,390 µg/L) for TBBPA ($n = 6$–8 per dose). For gene expression studies, exposure doses were $10^{-9}$ to $10^{-5}$ M ($n = 6$ per dose) for both compounds. Note that, due to the limited number of pituitaries, not all doses were included in all four cultures, leading the number of replicates to vary between treatments. Cultures were allowed to settle until day 4, by then the cell count was stable and the cells had started to spurt outgrowths, before being exposed to either BPA or TBBPA. Exposure was given as a single dose and incubation lasted 72 h.

### 2.4. Viability Assays

Viability testing was carried out using two non-toxic fluorescent indicator dyes, AlamarBlue (AB) and 5-carboxyfluorescein diacetate-actetoxymethyl ester (CFDA-AM) (both from Life Technologies). These assays indicate metabolic activity and plasma membrane integrity, respectively, and measure the conversion of a non-fluorescent dye into a fluorescent dye by enzymes present in intact and viable cells [46]. The test procedures followed the description by Hodne et al. [45]. In short, cells were seeded in 96-well plates (Corning, Amsterdam, The Netherlands) and incubated for 4 days before being exposed to either BPA or TBBPA for 72 h as described above. At day 7, culture medium in all wells was replaced with 100 µL Tris buffer (50 mM, pH 7.5) containing both 5% AB and 4 µM CFDA-AM (from 4 mM stock in dimethyl sulfoxide (DMSO)). After 30 min of incubation, the concentration of fluorescent products was measured simultaneously for both probes using a Bio-Tek FLX 800 fluorescence plate reader (Bio-Tek Instruments Inc., Winooski, VT, USA). Data was collected with Gen5 Data Analysis Software (Bio-Tek Instruments Inc.). As a positive control for cell toxicity, serving as intra/inter assay control, each plate included additional wells that were exposed to copper sulfate ($CuSO_4$; 0.156–2.5 mM, $n = 6$ per dose) for the last 24 h and analyzed alongside the experimental wells at day 7. The effects from $CuSO_4$ exposure were comparable between individual plates and cultures (data not shown) and similar to previously published data from this cell model (see von Krogh et al. [47], Supporting information), indicating stable cultures.

### 2.5. Quantification of Gene Expression

#### 2.5.1. RNA Extraction and cDNA Synthesis

Cells used for gene expression analysis were seeded in 24-well plates (Corning), exposed to either BPA or TBBPA at day 4 as described above, and harvested at day 7. Cells were lysed and homogenized by pipetting in Trizol (Life Technologies), from where total RNA was extracted, before being re-suspended in 10 µL RNase-free water (Ambion, Thermo Fisher Scientific, Waltham, MA, USA). DNase treated RNA (TURBO DNase-free (Ambion)) was quantified spectrophotometrically (NanoDrop, Thermo Fisher Scientific), and the quality assessed by electrophoretic validation (Bioanalyzer, Agilent

Technologies, Santa Clara, CA, USA) of the RNA Integrity Number (RIN). RNA samples with RIN numbers above 8 were allowed further analysis. Using 500 ng total RNA, first strand cDNA synthesis was performed using random hexamer primers and Super Script III (Life Technologies) according to standard procedures. cDNA was stored at −20 °C until qPCR.

### 2.5.2. Primers and Reference Genes

qPCR primers (Table 1) were designed using Primer3 shareware (http://frodo.wi.mit.edu/primer3/input.htm) and theoretically tested for possible hairpin loops and primer dimer formations using Vector NTI (Life Technologies). In each primer pair, one primer targeted an exon-exon border to avoid amplification of potential traces of genomic DNA. The genes of interest were *LHb* (GenBank ID: DQ402374), *FSHb* (GenBank ID: DQ402373), *GnRHr1b* (GenBank ID: GU332297) and *GnRHr2a* (GenBank ID: GU332298.1), all related to pituitary reproductive function. Four reference genes, *arp2*, *bactin*, *ubiquitin*, and *ef1a*, were tested using Bestkeeper Software [48], giving quantification cycle value (Cq), geometric means, and standard deviations of 27.06 (±0.39), 21.96 (± 0.45), 22.19 (±0.43), and 20.40 (±0.39), respectively. *ef1a* (GenBank ID: DQ402371.1) was considered most stably expressed and was subsequently used to normalize the qPCR data.

**Table 1.** qPCR primers used in the present study.

| Target | Reference | | Primer Sequence | Amplicon Size (nt) | Efficiency |
|---|---|---|---|---|---|
| *lhb* | [49] | Forward | 5′-GTGGAGAAGAAGGGCTGTCC-3′ | 81 | 1.93 |
| | | Reverse | 5′-GGACGGGTCCATGGTG-3′ | | |
| *fshb* | [49] | Forward | 5′-GAACCGAGTCCATCAACACC-3′ | 63 | 1.84 |
| | | Reverse | 5′-GGTCCATCGGGTCCTCCT-3′ | | |
| *gnrhr1b* | [3] | Forward | 5′-GCTACTCCCGAATCCTCCTC-3′ | 73 | 1.96 |
| | | Reverse | 5′-CGCCTCAGGTATGACTCTCC-3′ | | |
| *gnrhr2a* | [3] | Forward | 5′-TTCACCTTCTGCTGCCTCTT-3′ | 113 | 1.99 |
| | | Reverse | 5′-TCCGTGGAGGAAAGATTGTC-3′ | | |
| *bactin* | [45] | Forward | 5′-TTCTACAACGAGCTGAGAGTGG-3′ | 102 | 1.84 |
| | | Reverse | 5′-CATGATCTGGGTCATCTTCTCC-3′ | | |
| *arp2* | [45] | Forward | 5′-GGAGGTTAGAAGTAGCAAGGAGC-3′ | 107 | 1.94 |
| | | Reverse | 5′-TGCTGACTCTCACGGAGTTG-3′ | | |
| *ef1a* | [49] | Forward | 5′-CCTTCAACGCCCAGGTCAT-3′ | 100 | 1.92 |
| | | Reverse | 5′-AACTTGCAGGCGATGTGA-3′ | | |
| *ubiquitin* | [45] | Forward | 5′-TGTCAAAGCCAAGATTCAGG-3′ | 111 | 1.86 |
| | | Reverse | 5′-TGGATGTTGTAATCCGAGAGG-3′ | | |

### 2.5.3. qPCR Analysis

Using the LightCycler 480 platform (Roche, Basel, Switzerland), qPCR analyses were carried out as previously described [45,50]. Three non-template negative control (NTC) reactions were included for each primer pair by substituting the cDNA template with nuclease-free water (Ambion). To account for plate-to-plate variation, a standard positive calibrator control, prepared by mixing cDNA from all individual samples, was also included on every plate in triplicate. All samples were run in duplicate. Each PCR reaction (10 μL) mixture contained 5 μL of SYBR Green I master mix (Roche), 1 μl (5 μM) of forward primer, 1 μL (5 μM) of reverse primer, and 3 μl of diluted (1:10) cDNA. The qPCR reactions were carried out using an initial step for 10 min at 95 °C to activate the *Taq* polymerase, followed by 42 cycles consisting of 10 s at 95 °C (denaturation), 10 s at 60 °C (annealing), and elongation at 72 °C for 6 s. After every elongation, the fluorescence was measured and used to determine the Cq. Based on cDNA dilution curves, qPCR efficiencies (E) were determined. Combined with Cq values, E were used to calculate the relative expression [51,52] of each sample:

$$\text{Relative expression} = E_{\text{target}}{}^{\Delta Cq(\text{calibrator} - \text{sample})} * E_{\text{reference}}{}^{\Delta Cq(\text{sample} - \text{calibrator})}$$

Specificity of the qPCR assay was confirmed by agarose gel electrophoresis and amplicon sequencing at assay set-up, and by melting curve analysis based on slowly heating the reaction mixture from 65 °C to 98 °C directly following each individual PCR reaction.

*2.6. Statistical Analysis and Data Presentation*

Statistical analysis was performed using the JMP Pro14 software (SAS Institute Inc, Cary, NC, USA). As gene expression and cell viability occur at the level of the cell, and each culture well consisted of their own unique mixture of cells from multiple animals, individual wells were treated statistically as samples. Technical replicates from individual wells were averaged into one observation. Technical replicates had a variation of 0.26% (±0.69), while average variance between wells per treatment was 17.1% (±9.63). Fold changes of exposed samples relative to their respective solvent control mean were calculated and used in the subsequent analysis for both gene expression and viability data. To maintain control variance in the analyses, control samples were calculated in the same manner. All data were tested for normality and equal variances by Shapiro–Wilk W test and Levene's test, respectively. If the criteria for parametric testing was met, a one-way ANOVA was performed, followed by Dunnett's method comparing individual groups to the solvent control if these were found to have originated from different populations. However, most data did not meet the assumptions of parametric testing, even after transformation, so non-parametric testing was used. To assess population differences, the Wilcoxon rank-sum test was used to compare solvent control with control blank, whereas the Kruskal–Wallis test was used to compare treatment groups. In the latter case, if the null hypothesis was rejected, statistical differences between treatment groups and the solvent control were assessed by the Steel method. The significance level for rejection of the null-hypothesis was set to 0.05. Scatter plots were made using Prism 8 (GraphPad Software, San Diego, CA, USA). In each graph, the mean ± standard error of the mean (SEM) is indicated. Outliers of more than 2-fold difference to the group mean were excluded from the dataset, while all outliers were included if the statistical test was non-parametric.

## 3. Results

*3.1. Solvent Control vs Control Blank*

For every culture prepared, control wells with and without 0.2% EtOH (solvent control and control blank, respectively) added to the media were included. The AB and CFDA-AM viability assays performed on these cells revealed that the solvent had statistically significant negative effects on both metabolic activity and membrane integrity (Supplementary data, Figure S1). No influence on gene expression was detected from the solvent (Supplementary data, Figure S2). Note that data from exposed cells in the following sections are compared to the effects seen in solvent control cells.

*3.2. BPA Exposure*

BPA at all experimental doses ($10^{-9}$ M to $10^{-3}$ M) affected cell metabolic activity (Figure 1A), while all except $10^{-9}$ M affected membrane integrity (Figure 1B). A hormesis effect was detected in both assays, with low doses stimulating and high doses inhibiting cell viability. At $10^{-3}$ M, no membrane integrity was detected. In the AB assay, lower BPA concentrations increased data variance compared to higher concentrations. At $10^{-4}$ and $10^{-3}$ M, the cellular outgrowths visible at lower concentrations and control had disappeared completely (see example in Supplementary data, Figure S3).

Gene expression levels of gonadotropin subunits *FSHb* and *LHb* (Figure 2A,B, respectively), and of GnRH receptor *GnRHr1b* (Figure 3A), were unaffected by BPA exposure at all doses tested ($10^{-9}$ M to $10^{-5}$ M). In contrast, doses from $10^{-7}$ M to $10^{-5}$ M increased *GnRHr2a* transcription levels (Figure 3B).

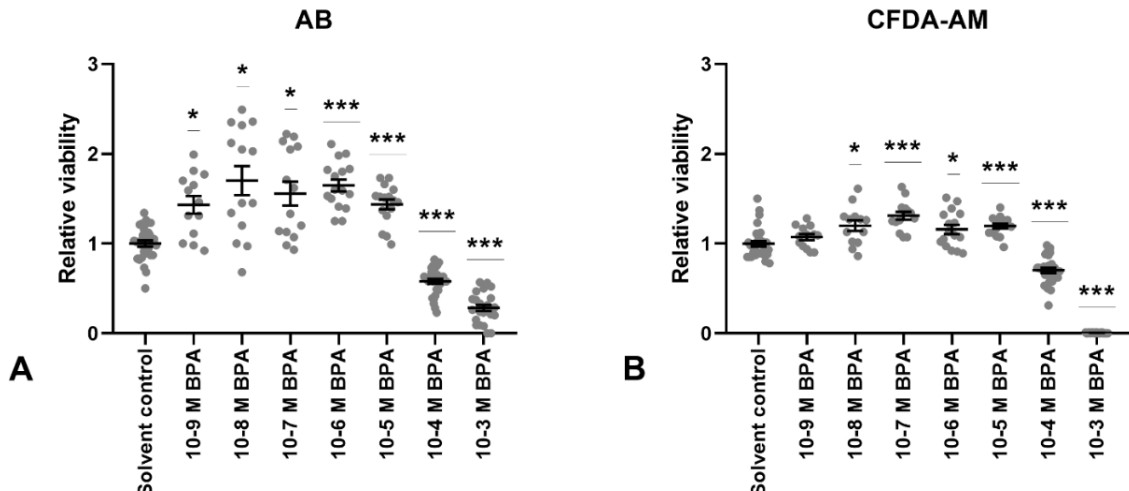

**Figure 1.** Viability in terms of metabolic activity (AlamarBlue (AB) assay) (**A**) and membrane integrity (5-carboxyfluorescein diacetate-actetoxymethyl ester (CFDA-AM) assay) (**B**) in Atlantic cod pituitary cells after 7 days of primary culture w/wo Bisphenol A (BPA) added to the culture media for the last 72 h. Each data point represents the sample fold change relative to mean solvent control. For each column, the group mean ± SEM is indicated ($n$ = 13–30). Statistical significance (* $p < 0.05$, *** $p < 000.1$) was assessed using the Kruskal–Wallis test followed by the Steel method.

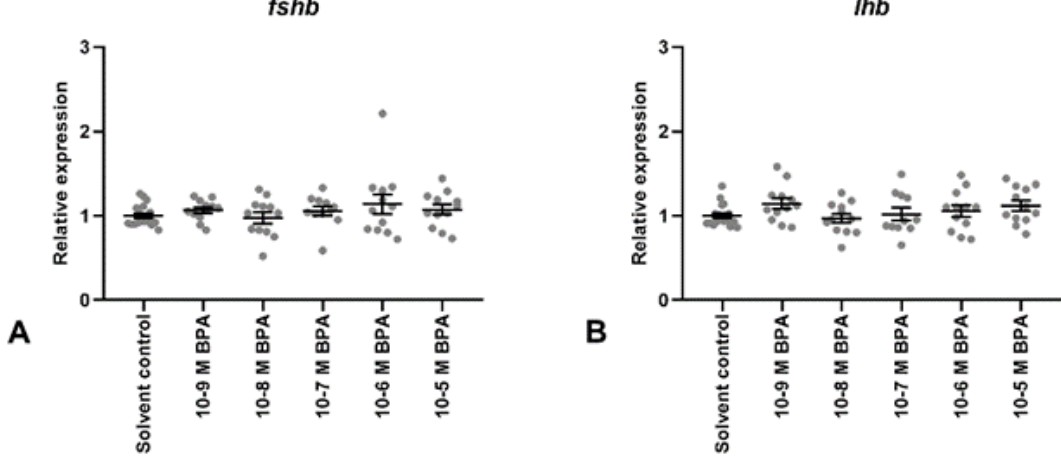

**Figure 2.** Gene expression of gonadotropin subunits (*FSHb*; (**A**), *LHb*; (**B**)) in Atlantic cod pituitary cells after 7 days of primary culture w/wo BPA added to the culture media for the last 72 h. Each data point represents the sample fold change relative to mean solvent control. For each column, the group mean ± SEM is indicated ($n$ = 11–18). No statistical differences between groups were detected (tested by Kruskal–Wallis; (**A**), or one-way ANOVA; (**B**)).

### 3.3. TBBPA Exposure

All experimental doses of TBBPA ($10^{-9}$ M to $10^{-4}$ M) significantly affected cell viability, both in terms of metabolic activity and membrane integrity (Figure 4A,B, respectively). As with BPA, a hormesis effect was detected for both parameters. TBBPA cytotoxicity appeared ten times more potent than that of BPA on these cells, as negative effects were detected from $10^{-5}$ M TBBPA. At $10^{-4}$ M, no membrane integrity was detected. At $10^{-5}$ and $10^{-4}$ M, no cellular outgrowths were visible through the microscope.

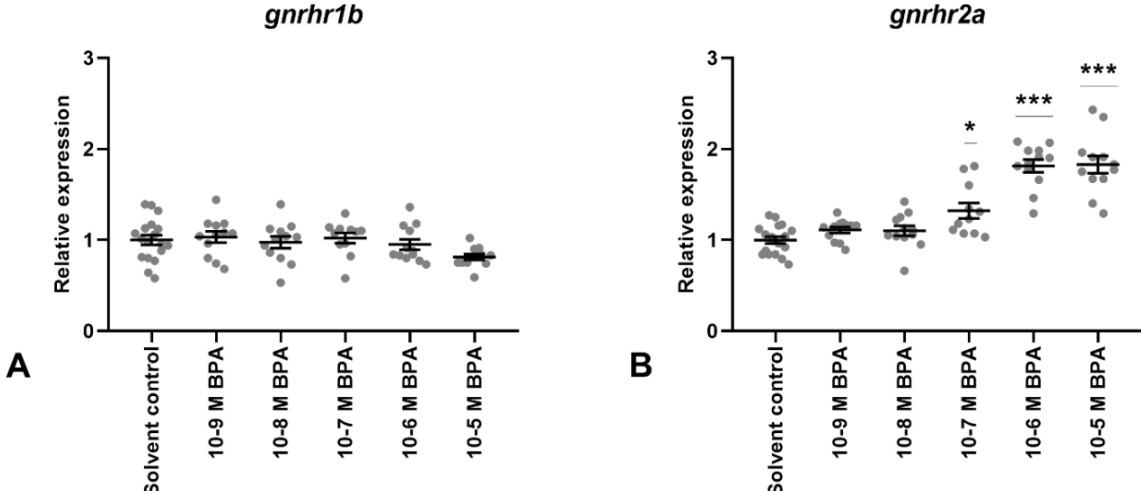

**Figure 3.** Gene expression of two gonadotropin-releasing hormone receptors (*GnRHr1b*; (**A**), *GnRHr2a*; (**B**)) in Atlantic cod pituitary cells after 7 days of primary culture w/wo BPA added to the culture media for the last 72 h. Each data point represents the sample fold change relative to mean solvent control. For each column, the group mean ± SEM is indicated (*n* = 11–18). Statistical significance (* *p* < 0.05, *** *p* < 000.1) was assessed using a one-way ANOVA followed by Dunnett's method.

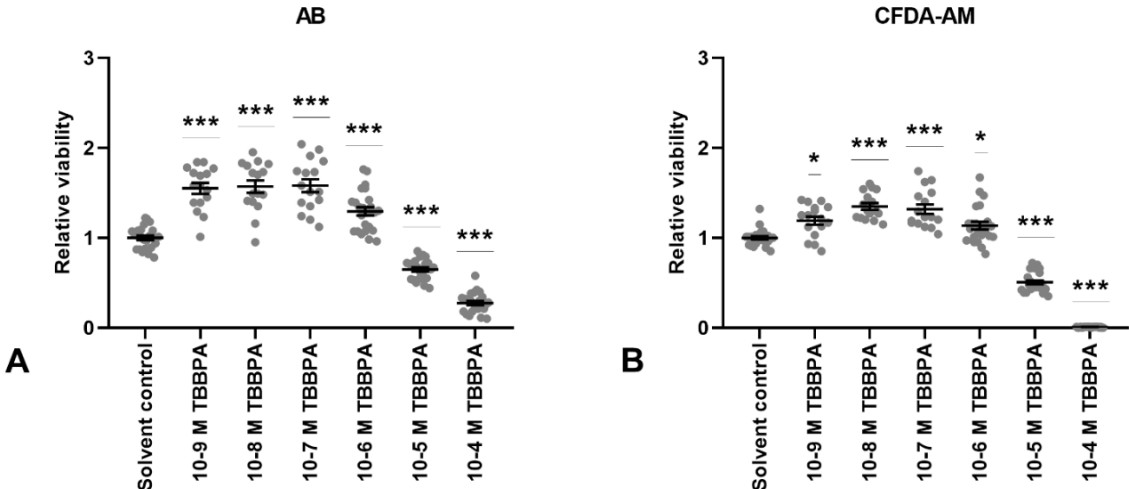

**Figure 4.** Viability in terms of metabolic activity (AB assay; (**A**)) and membrane integrity (CFDA-AM assay; (**B**)) in Atlantic cod pituitary cells after 7 days of primary culture w/wo tetrabromobisphenol A (TBBPA) added to the culture media for the last 72 h. Each data point represents the sample fold change relative to mean solvent control. For each column, the group mean ± SEM is indicated (*n* = 16–24). Statistical significance (* *p* < 0.05, *** *p* < 000.1) was assessed using the Kruskal–Wallis test followed by the Steel method.

Small, but statistically significant, increases in *FSHb* expression levels (Figure 5A) were observed in samples treated with $10^{-9}$ M to $10^{-7}$ M TBBPA. Similarly, a non-significant tendency was seen after $10^{-6}$ M exposure (*p* = 0.11). Levels of *LHb* mRNA (Figure 5B) increased significantly after $10^{-9}$, $10^{-8}$, and $10^{-6}$ M treatment, while a small, non-significant, increase was observed in $10^{-7}$ M (*p* = 0.17) samples. No changes were detected after $10^{-5}$ M TBBPA exposure.

Gene expression levels of both GnRH receptors (*GnRHr1b* and *grnrh2a*; Figure 6A,B, respectively) was unaffected by TBBPA concentrations of $10^{-9}$ M to $10^{-6}$ M, but decreased significantly after $10^{-5}$ M TBBPA treatment. In terms of gene expression, TBBPA exposed samples displayed larger in-group variation than BPA exposed samples.

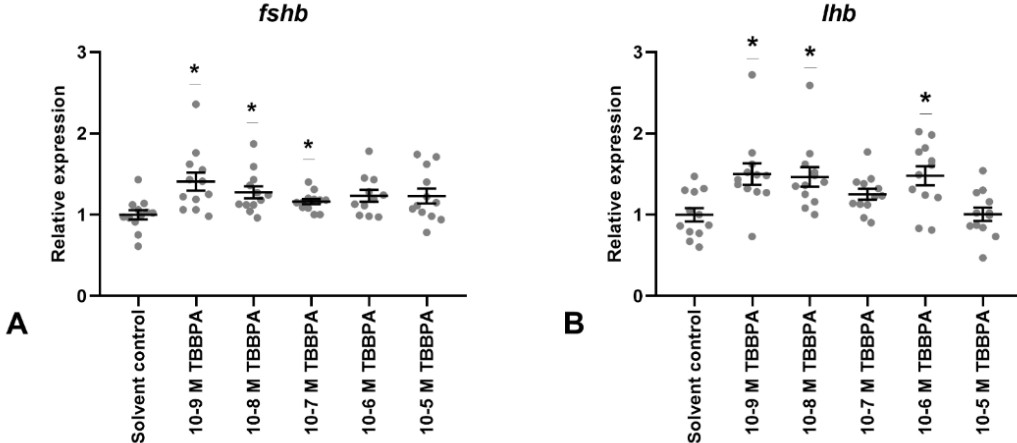

**Figure 5.** Gene expression of gonadotropin subunits (*FSHb*; (**A**), *LHb*; (**B**)) in Atlantic cod pituitary cells after 7 days of primary culture w/wo TBBPA added to the culture media for the last 72 h. Each data point represents the sample fold change relative to mean solvent control. For each column, the group mean ± SEM is indicated (*n* = 11–12). Statistical significance (* $p < 0.05$) was assessed using the Kruskal–Wallis test followed by the Steel method.

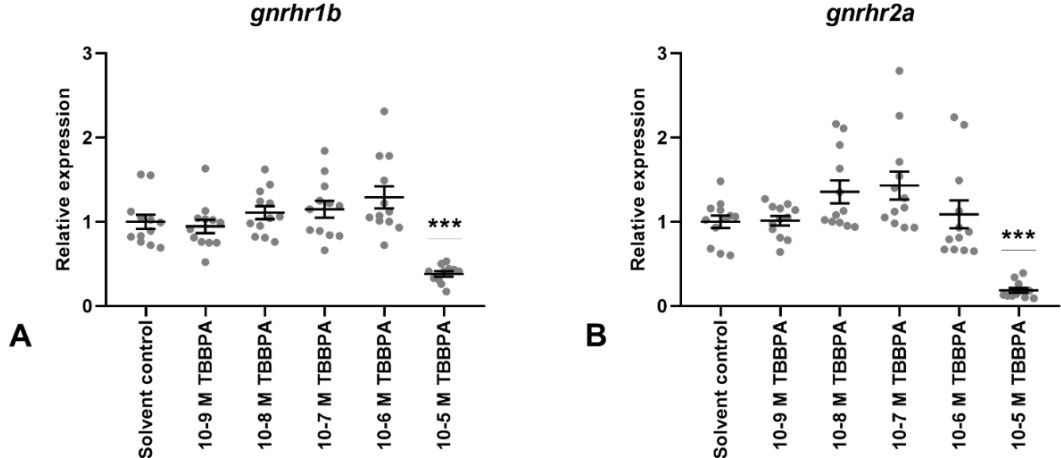

**Figure 6.** Gene expression of two gonadotropin-releasing hormone receptors (*GnRHr1b*; (**A**), *GnRHr2a*; (**B**)) in Atlantic cod pituitary cells after 7 days of primary culture w/wo TBBPA added to the culture media for the last 72 h. Each data point represents the sample fold change relative to mean solvent control. For each column, the group mean ± SEM is indicated (*n* = 11–12). Statistical significance (*** $p < 000.1$) was assessed using the Kruskal–Wallis test followed by the Steel method.

## 4. Discussion

All wildlife is potentially susceptible to the impacts of anthropogenic activity. The substantial use and subsequent environmental release of synthetic compounds are examples of such activity and a source of increasing concern. This study demonstrates that single, acute exposure to either BPA or TBBPA at environmentally relevant concentrations can affect cell viability and reproduction-related gene expression in Atlantic cod pituitary cells, potentially affecting the reproductive capacities of this species.

### 4.1. Methodological Considerations

To obtain physiologically relevant results using in vitro systems for toxicological testing, it is crucial that cells are kept under physiologically relevant conditions [45,53]. Nevertheless, it is common practice to use mammalian protocols on fish tissue cultures, with only the incubation temperature adjusted to accommodate differences between mammalian and fish physiology. The present study

followed protocols optimized for Atlantic cod physiology, regarding not only temperature, but also cell density, $pCO_2$, and pH and osmolality of the working solutions and incubation media [45]. These optimized conditions allow cultured pituitary cells to maintain stable membrane potentials, fire action potentials, and to exhibit steady GnRH responses for at least two weeks. Moreover, cell viability is significantly improved compared to traditional cell culture conditions [45].

To be able to expose the pituitary cells to high concentrations of BPA and TBBPA, the bisphenols were initially dissolved in EtOH. The working concentration of 0.2% did not affect gene expression of any gene analyzed, but did negatively affect cell viability compared to those incubated in EtOH free media (control blank). Consequently, all exposed cells were compared to that of the solvent control. Most studies use either EtOH or dimethyl sulfoxide (DMSO) as the solvent for BPA and TBBPA, both of which are able to induce cytotoxicity [54,55]. Despite this, effects from the solvent itself are rarely reported, making comparison between studies difficult in this regard. For future studies, we would recommend lower EtOH working concentrations, as negative effects were detected here. The water solubility of BPA and TBBPA is 120–300 mg/L and 4.16 mg/L at 25 °C, respectively, with decreasing solubility at lower temperatures [5,15]. The cell medium used for the cod cultures was prepared and incubated at 12 °C. The highest doses administered to the cells thus exceed the water solubility threshold for these compounds and are unlikely to be environmentally relevant for Atlantic cod. They may still, however, be mechanistically informative. It should also be noted that high concentrations, similar to the higher doses administered in this study, have been detected in landfill leachates (17.2 mg/L for BPA) [9] and freshwater sediment (9.8 mg/kg DW for TBBPA) [10], and could potentially reach other wildlife. In marine waters, levels are usually below 1 µg/L (4.39 nM) for BPA and below 1.8 µg/L (3.3 nM) for TBBPA [7,12,16], which corresponds roughly to the two lowest doses, $10^{-9}$ and $10^{-8}$ M (0.23 and 2.28 µg/L for BPA, and 0.54 and 5.44 µg/L for TBBPA, respectively), used in this study. In Atlantic cod liver, detected levels of BPA and TBBPA have been 107.2 and 9 ng/g lipid weight (LW), respectively [56].

Cells were exposed once (at 4 day post plating) and incubated for 72 h prior to harvest and analysis. While the half-life of BPA and TBBPA in fish is rather short, <1 day [5,15], in a pituitary cell culture, the metabolism is expected to be slower than in the intact body, as the main site of metabolism is in the liver. The final concentrations of the compounds or possible metabolites formed during the incubation were not measured. Metabolites may have differential properties than the parent compound, and we cannot rule out the possibility that any observed effect described here is a result of such.

*4.2. BPA*

BPA, often referred to as a xenoestrogen because of its ability to act as an estrogen agonist, binds to estrogen receptors (ERS) and promotes endogenous 17β-estradiol (E2) effects in vertebrates [9]. Not surprisingly, the vast majority of reproductive studies on BPA in fish has been on ovaries or hepatic vitellogenin production, which are important sites of estrogenic activity. The studies reveal that in many teleosts, including Atlantic cod, BPA is able to induce vitellogenin production and act as an estrogen [35,37,57,58]. In addition to the gonads and liver, estrogen acts on the higher levels of the reproductive axis. Correspondingly, in the brain of developing zebrafish (*Danio rerio*) larvae, BPA has been shown to activate estrogenic markers, such as enhanced transcription of brain aromatase (*cyp19a1b*), through Er-dependent transcription [19,21]. In a recent study using Atlantic cod pituitary cultures, 72 h of 25 ng/mL E2 exposure had no effect on gene expression levels (*FSHb*, *LHb*, *GnRHr1b*, *GnRHr2a*) in cells from sexually maturing donors [3]. In the current study, BPA had no effect on *FSHb*, *LHb*, or *GnRHr1b* at any concentration tested but was able to stimulate *GnRHr2a* expression at $10^{-7}$ to $10^{-5}$ M (22.8–2282.9 µg/L), the lowest dose being similar to the E2 dose previously used by von Krogh et al. [3]. This indicates that BPA is able to induce *GnRHr2a* expression through some Er-independent pathway. While E2 did not induce gene expression in maturing Atlantic cod pituitary cells, it stimulated the expression of both *FSHb* and *GnRHr2a* in cells from sexually mature fish [3].

Keeping this is mind, it is possible that BPA exposure at different stages of the reproductive cycle could, like E2, induce differential effects.

Another physiological aspect of E2 is the ability to stimulate cell proliferation, including at the pituitary level [59]. In sexually maturing Atlantic cod pituitary cultures, 25 ng/mL, but not 2.5 ng/mL, E2 stimulated cell viability [3]. Here, however, BPA was able to stimulate viability at lower concentrations than did E2, increasing metabolic activity from $10^{-9}$ M and membrane integrity from $10^{-8}$ M. Similar proliferative potency has been seen in the offspring of female mice, with increased pituitary proliferation and gonadotrope numbers when exposed prenatally to just 0.5 µg/kg/day of BPA [60]. In concurrence with concentrations considered safe by authorities (Predicted no effect concentration (PNEC) for marine waters; 0.15 µg/L) [6], no adverse effects were detected after BPA exposure at this level. However, metabolic activity was stimulated by concentrations close to the safe limit (i.e., $10^{-9}$ M/0.228 µg/L), indicating that cellular mechanisms may be affected even at this level. Both through visual inspection and cell viability testing, doses of $\geq 10^{-4}$ M were found to induce cytotoxicity, and at $10^{-3}$ M, no membrane integrity was detected. Similar results have been measured in a zebrafish hepatocyte cell line (ZFL), with 24h $LC_{50}$ and 96h $LC_{50}$ for BPA of 367.1 and 357.6 µM, respectively [61]. In chicken embryonic hepatocytes, the 36h $LC_{50}$ for BPA was somewhat lower, calculated at 61.7 µM [62].

In the present study, BPA exposure did not affect transcript levels of *FSHb*, *LHb*, or *GnRHr1b* at any dose but increased *GnRHr2a* expression at $\geq 10^{-7}$ M. To our knowledge, there are no other studies looking at this in vitro, making direct comparison difficult. There are, however, some studies in vivo. The first teleost study looking at the effect of BPA exposure on gonadotropins was conducted by Rhee et al. [22], using the mangrove killifish (*Kryptolebias marmoratus*) as the model. Waterborne BPA exposure increased transcript levels of the gonadotropin β-subunits in both juvenile (300 µg/L for 24 h) and adult fish (600 µg/L for 96 h). Similar results have been reported from zebrafish. Qiu et al. [23] investigated BPA exposure during development (from 2 to 120 hpf) in zebrafish larva and found increased mRNA levels of *FSHb* after 1000 µg/L and *LHb* after 10, 100, and 1000 µg/L treatments. In juvenile zebrafish, exposure to $10^{-5}$ M BPA for 20 days increased *LHb*, but decreased *FSHb* pituitary expression [24]. In contrast, the recent work of Wang et al. [25], showed that female goldfish (*Carassius auratus*) exposed to 1, 50, and 500 µg/L BPA for 30 days had reduced levels of *FSHb* and *LHb* transcripts, whereas no effect was detected in males. The discrepancies between the studies described above and also our own results could be due to species-specific differences, for instance in BPA Er affinity, uptake, or metabolism. Furthermore, experimental design (e.g., in vivo vs in vitro), age of the experimental animals, and exposure time could all affect the outcome. Conflicting results have also been reported on the effects of BPA in mammalian studies. Results are conflicting, also in mammalian reports. For instance, rats treated postnatally from day 21–35 with 2.4 µg/kg/day had decreased serum LH and pituitary *LHb* levels [63], while adult male rats exposed to 5 and 25 mg/kg/d for 40 days, had reduced serum FSH and LH, but increased *FSHb* and *LHb* transcript levels [64]. Moreover, female mice treated with BPA at 0.5 µg/kg/day had increased gonadotropin mRNA levels, whereas mice treated with 50 µg/kg/day had decreased levels [60].

The GnRH receptors mediate hypothalamic GnRH responses on the pituitary cells, and in Atlantic cod, *GnRHr2a* is thought to be the main modulator of gonadotropin regulation [65]. In Atlantic cod, expression of *GnRHr1b* and *GnRHr2a* are susceptible to glucocorticoid regulation, and the latter also to sex steroid regulation in Atlantic cod [3,47], making them both likely targets for endocrine disruption. Presently, *GnRHr1b* expression was unaffected by BPA, while *GnRHr2a* transcription increased following exposure to $10^{-7}$, $10^{-6}$, and $10^{-5}$ M BPA. Acute exposure to $\geq 0.1$ µM BPA could thereby, assuming translation into protein, increase *GnRHr2a* receptor levels, enhancing GnRH sensitivity and subsequent LH production. Also in this regard, there are no in vitro studies for comparison. However, in the killifish (*K. marmoratus*), in vivo exposure to 600 µg/L BPA for 96 h increased brain/pituitary transcript levels of *GnRHr*, an orthologue of Atlantic cod *GnRHr2a*, in secondary males, similarly to our findings [66]. In contrast, no effect was seen in hermaphrodite killifish. Li et al. [67] found no effect on brain *GnRHr*

(also a *GnRHr2a* orthologue) expression from 1, 10, 50, 125, and 250 mg/kg BPA exposure in tongue sole (*Cynoglossus semilaevis*) females, whereas a decrease was seen in males exposed to 250 mg/kg BPA. While the present study did not detect any effects from BPA on *GnRHr1b* levels, the orthologue *GnRHr1a,* was significantly upregulated in brains of adult female rare minnows (*Gobiocypris rarus*) following 15 μg/L BPA exposure for 35 days [17]. No effect was detected in males. The above mentioned studies describe brain levels, or in one case a mixture between brain and pituitary levels, of *GnRHr*, and we have not managed to find a teleost study looking solely at pituitary *GnRHr* levels. In mammals, however, pituitary *GnRHr*, the paralogue to teleost *GnRHr*, showed increased transcript levels after 40 days of 5 and 25 mg BPA/kg/d in adult male Wistar rats [64]. In female mice pituitaries, *GnRHr* transcripts increased after treatment with 0.5 μg/kg/day BPA, but decreased after 50 μg/kg/day treatment [60]. Although little information is available, it appears as if there are both species and gender differences regarding the effect of BPA on *GnRHr* gene expression. In the present study, due to a limited number of available fish and lack of external dimorphic traits in Atlantic cod, the pituitary cultures were prepared as a mixture between male and female donors. For future mechanistic studies, it seems to be worth pursuing cultures separated by sex.

## 4.3. TBBPA

The effects of TBBPA on fish reproduction is far less studied than that of BPA. Though studies have indicated that TBBPA can affect teleost reproductive success [28,30,31], there are no previous studies, known to us, looking at the higher levels of the BPG-axis in this animal group. Here, we demonstrate that TBBPA can stimulate pituitary cell viability and gene expression of gonadotropin β-subunits at acute, environmentally relevant concentrations.

In the present study, $10^{-9}$ to $10^{-6}$ M TBBPA exposure stimulated both metabolic activity and membrane integrity in cod pituitary cells. At some doses, the cell viability levels of TBBPA exposed cells exceeded the level of not only the solvent control, but also the control blank samples, most likely reflecting cell proliferation. Similar findings, albeit at higher concentrations, were previously demonstrated in a rat pituitary tumor cell line, GH3, where $10^{-6}$ to $10^{-4}$ M TBBPA induced cell growth and increased growth hormone (GH) production [41]. While E2 may be a stimulator of proliferation, as mentioned above, the estrogenic properties of TBBPA is still debated. Nevertheless, in the human breast adenocarcinoma cell line, MCF-7, estrogenic proliferation was seen following 24 h TBBPA exposure up to 500 μM [34]. This is a much higher concentration than that which induced cytotoxicity in cod pituitary cells, $\geq 10^{-5}$ M (10 μM), indicating differential sensitivity between cell types. However, our findings are in concert with other teleost studies. For instance, in ZFL cells, the 24 h $LC_{50}$ and 96 h $LC_{50}$ was 4.0 and 4.2 μM, respectively [61]. The zebrafish standard embryo assay demonstrated that TBBPA $\geq$ 0.75 mg/L (1.38 μM) caused lethality or malformation [68]. Similar results are reported for zebrafish larvae, where the 96 h $LC_{50}$ for TBBPA was 5.27 mg/L (9.7 μM) [37]. Many mammalian studies also report similar findings. In GH3 cells, measuring the reduction of blue resazurin dye to red fluorescent resorufin, the same cytochrome as used in the AlamarBlue assay, cytotoxicity was detected at doses >1 μM [33]. Moreover, in primary cultures of rat cerebellar granule cells, trypan blue measurements, indicating loss of membrane integrity in stained cells, demonstrated a 24 h $LC_{50}$ value of 7 μM TBBPA [69]. The same study demonstrated that all membrane integrity was lost at 20 μM TBBPA, in concert with the present results.

The cytotoxicity of EtOH was apparently counteracted by both bisphenols tested in the present study. However, the EtOH reduced cell viability was not reflected in reduced gene expression compared to control blank cells. Similarly, the cytotoxicity induced by $10^{-5}$ M TBBPA exposure did not affect *FSHb* and *LHb* expression, which both remained at the level of the solvent control. The lowered cell viability did, however, correlate with reduced gene expression of the *GnRHr* receptors that were otherwise unaffected by TBBPA treatment. It is not clear to us what might cause this discrepancy and further studies are needed for elucidation.

As mentioned above, studies on the potential effects of TBBPA at the higher levels of the BPG axis are currently very limited. To our knowledge, there are only two vertebrate studies, only one of which concerns gonadotropin levels. Further, to our knowledge, no study has investigated the effect of TBBPA exposure on GnRH receptor expression or possible direct effects at the pituitary level. van der Ven et al. [44] found no obvious changes in pituitary, nor gonadotrope, histology after TBBPA exposure in a one-generation study using Wistar rats. However, male pituitary weight dose-dependently increased. The authors could not conclude the mechanisms behind this, but suggested that it might be caused by feedback stimulation by the observed decreased levels of circulating thyroxine (T4). In a recent study by Zhang et al. [43], adult male frogs (*Rana nigromaculata*) were exposed to 0.001–1 mg/L TBBPA for 14 days. Doses 0.001–0.1 mg/L decreased serum LH, while 0.1 and 1 mg/L decreased serum FSH. This is in the opposite direction of our findings, where TBBPA stimulated gene expression of both gonadotropins, even at environmentally relevant concentrations. On the other hand, TBBPA treatment also led to increased serum testosterone and E2 levels in the male frogs. It is therefore possible that the reduced FSH and LH levels were a result of negative feedback from the sex steroids at the brain and pituitary level, rather than direct effects from TBBPA. It is also possible that the steroid action masked potential direct effects from TBBPA at the pituitary level. While direct effects on gonadotropin expression are evident from the present findings, it remains to be seen if this could lead to physiological alterations in vivo for Atlantic cod.

The mechanisms at which TBBPA affects gene expression and other reproductive endpoints remains to be assessed. However, TBBPA has the potential to act as a thyroid hormone agonist [41], and disruption of the thyroid axis can lead to sexual dysfunction [70]. Furthermore, thyroid hormones are thought to be involved in the seasonal regulation of reproduction [71]. It would therefore be interesting to investigate possible TBBPA effects throughout the reproductive cycle. Not only for the connection to thyroid hormones and because available information on the reproductive effects of TBBPA is limited, but also because the sensitivity of potential endocrine targets might differ through the different stages of sexual maturity [3,47].

## 5. Conclusions

The present study demonstrates that single, acute, and environmentally relevant doses of BPA or TBBPA are able to directly affect both cell viability and gene expression in Atlantic cod dispersed pituitary cells in vitro. Cell viability was stimulated by low dose exposure to both phenols, while cytotoxicity was evident only at high, non-environmentally relevant doses. Bromination seemingly increased cytotoxicity, as TBBPA exposure decreased pituitary cell viability at 10-fold lower doses than BPA. BPA stimulated gene expression of the GnRH receptor 2a, the main gonadotropin modulator in Atlantic cod, while TBBPA stimulated gene expression of both gonadotropins. The differential effects seen from BPA and TBBPA exposure at the transcriptional level indicate that they act on different receptors or activate different pathways.

**Supplementary Materials:** The following are available online at http://www.mdpi.com/2410-3888/4/3/48/s1, Figure S1: Viability assays, controls, Figure S2: Gene expression, controls, Figure S3: Morphological changes, control cells vs high BPA exposure cells.

**Author Contributions:** Conceptualization, K.v.K., E.R., T.M.H., and F.-A.W.; data curation, K.v.K. and R.N.-L.; formal analysis, K.v.K.; funding acquisition, T.M.H. and F.-A.W.; investigation, K.v.K.; methodology, K.v.K., R.N.-L., and T.M.H.; project administration, F.-A.W.; resources, E.R. and F.-A.W.; supervision, E.R., T.M.H., and F.-A.W.; validation, K.v.K., E.R., R.N.-L., T.M.H., and F.-A.W.; visualization, K.v.K.; writing—original draft, K.v.K.; writing—review and editing, E.R., R.N.-L., T.M.H., and F.-A.W.

**Funding:** This research received no external funding.

**Acknowledgments:** The authors would like to thank Ketil Hylland, Ørjan Karlsen, and the crew at R/V Trygve Braarud for help aquiring cod. We kindly thank Ian Mayer for proofreading the manuscript. Our appreciation goes also to Gersende Maugars for phylogeny assistance. This research was fundedby the Research Council of Norway (Grant Nos. 184851 and 191825) and by the Norwegian University of Life Sciences.

**Conflicts of Interest:** The authors declare that there is no conflict of interest that could be perceived as prejudicing the impartiality of the research reported.

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
