# Peer review of "In Vitro Effects of Bisphenol A and Tetrabromobisphenol A on Cell Viability and Reproduction-Related Gene Expression in Pituitaries from Sexually Maturing Atlantic Cod (Gadus morhua L.)"

_fishes, doi:10.3390/fishes4030048_

Round 1

Reviewer 1 Report

Thank you very much for giving me the opportunity to review the article entitled "In vitro effects of BPA and TBBPA on cell viability and 
reproduction-related gene expression in pituitaries from sexually maturing 
Atlantic cod (Gadus morhua L.)" by von Krogh et al. The paper is interesting, but I have a couple of methodological considerations that prevent me from accepting it in its present form:

Major:

1-In the methods part, you state that the number of cultures is 4, so how is it possible to have n=13-30 in the viability assay? Are you counting individual wells as a different sample? Different wells from a single culture would be technical replicates, if they come from a pool of tissue from different animals (page 6, line 210).

2-Same as before, for the gene expression assays, you state that n=11-18 (page 6, line 220). Please explain.

Minor:

Revise English grammar throughout the introduction, for example "One important and susceptible endocrine system is that of reproduction.", page 1 line 40 should say: One important and susceptible endocrine system is the reproductive axis.

Author Response

Reviewer 1

The authors would like to thank Reviewer 1 for taking the time to review our manuscript. Please find our responses in italic below.

Major:

1-In the methods part, you state that the number of cultures is 4, so how is it possible to have n=13-30 in the viability assay? Are you counting individual wells as a different sample? Different wells from a single culture would be technical replicates, if they come from a pool of tissue from different animals (page 6, line 210).

Kindly see below, as we address this comment together with comment 2.

2-Same as before, for the gene expression assays, you state that n=11-18 (page 6, line 220). Please explain.

The n reflects total number of wells from four separate cultures combined. In events where several measurements were performed on individual wells, like qPCR replicates, these were averaged into one observation.

While you are correct that different wells from a single culture can represent a form of technical replication, this is highly dependent on which cells are used. Using cell-lines, where all cells essentially are clones, individual wells are to be regarded technical replicates. However, primary cultures combining tissue from several animals will contain different cell types and different genetic profiles. Pooling the pituitaries and then randomly distributing the cells into individual wells will cause each well to consist of its own unique mixture of cells from different animals. As gene expression and cell viability occur at the level of the cell, and not the organism, biological variation between the wells are ensured.

Making separate cultures for each fish was not an option as one pituitary does not contain enough cells for all controls and treatments. Pooling pituitaries was therefore necessary. While pooling pituitaries gives more cells, it does reduce variability between wells compared to wells prepared from individual animals. However, using wild fish gives more variability than fish reared in fish facilities with high genetic similarity and uniform environment and life histories. The biological difference between wells in the present study was evident in the gene expression analysis where technical replicates from single wells differed by a maximum of 4% (average 0.26% ±0.69), while average variation between wells per treatment was 17.1% (±9.63). 

We have clarified this in the methods, new line 209-2013

Minor:

Revise English grammar throughout the introduction, for example "One important and susceptible endocrine system is that of reproduction.", page 1 line 40 should say: One important and susceptible endocrine system is the reproductive axis.

We consulted a native English speaker and fish biologist professor to proofread the manuscript. Corrections are made visible using “Track changes”. 

Reviewer 2 Report

The current manuscript has been written very well. I would recommend publication with minor corrections.

Minor Comments:

Please include references after the following sentences in the introduction section.

“Bisphenol A (BPA) and its halogenated derivative tetrabromobisphenol A (TBBPA) are widely 48 used industrial and commercial chemicals, the former mainly in production of polycarbonate plastics 49 and epoxy resins, and the latter as a flame retardant in combustible products”

“Both bisphenols are 50 ubiquitously detected in the environment, although at highly variable concentrations dependent on 51 sample matrix and location.”

Under RNA Extraction (Materials and Methods: Please mention from where trizol was purchased (Life Technologies, USA????)

Author Response

Reviewer 2

The authors would like to thank Reviewer 2 for taking the time to review our manuscript. Please find our responses in italic below.

Minor Comments:

Please include references after the following sentences in the introduction section.

“Bisphenol A (BPA) and its halogenated derivative tetrabromobisphenol A (TBBPA) are widely used industrial and commercial chemicals, the former mainly in production of polycarbonate plastics and epoxy resins, and the latter as a flame retardant in combustible products”

Additional references added to line 65

“Both bisphenols are ubiquitously detected in the environment, although at highly variable concentrations dependent on sample matrix and location.”

Additional references added to line 66

Under RNA Extraction (Materials and Methods: Please mention from where trizol was purchased (Life Technologies, USA????) 

Throughout the manuscript, we have made the choice to include headquarters and country of manufacturers only at their first mention. For instance, Life Technologies is first described in line 105. There, it is referenced as “Life Technologies, Carlsbad, CA, USA”, whereas later in the manuscript, we refer to the manufacturer by company name alone.
